# Nasal Polyposis Quality of Life (NPQ): Development and Validation of the First Specific Quality of Life Questionnaire for Chronic Rhinosinusitis with Nasal Polyps

**DOI:** 10.3390/healthcare10020253

**Published:** 2022-01-28

**Authors:** Ilaria Baiardini, Giovanni Paoletti, Alessia Mariani, Luca Malvezzi, Francesca Pirola, Giuseppe Spriano, Giuseppe Mercante, Francesca Puggioni, Francesca Racca, Giulio Melone, Giacomo Malipiero, Sebastian Ferri, Giorgio Walter Canonica, Enrico Heffler

**Affiliations:** 1Personalized Medicine Asthma and Allergy, Humanitas Clinical and Research Center, IRCCS, Rozzano, 20089 Milan, Italy; ilaria.baiardini@libero.it (I.B.); francesca.puggioni@humanitas.it (F.P.); francesca.racca@humanitas.it (F.R.); giacomo.malipiero@gmail.com (G.M.); sebastian.ferri@humanitas.it (S.F.); giorgio_walter.canonica@hunimed.eu (G.W.C.); enrico.heffler@hunimed.eu (E.H.); 2Department of Biomedical Sciences, Humanitas University, Pieve Emanuele, 20089 Milan, Italy; alessiamariani1995@gmail.com (A.M.); giuseppe.spriano@hunimed.eu (G.S.); giuseppe.mercante@hunimed.eu (G.M.); giuliomelone11@gmail.com (G.M.); 3Otorhinolaryngology Unit, Humanitas Clinical and Research Center, IRCCS, Rozzano, 20089 Milan, Italy; luca.malvezzi@humanitas.it (L.M.); francesca.pirola@humanitas.it (F.P.)

**Keywords:** chronic rhinosinusitis, nasal polyps, patient reported outcomes, quality of life, validation

## Abstract

To date, no disease-specific tool has been available to assess the impact of chronic rhinosinusitis with nasal polyps (CRSwNP) on health-related quality of life (HRQoL). Therefore, the purpose of this study was to develop and validate a questionnaire specifically designed to this aim: the Nasal Polyposis Quality of Life (NPQ) questionnaire. As indicated in the current guidelines, the development and validation of the NPQ occurred in two separate steps involving different groups of patients. The questionnaire was validated by assessing internal structure, consistency, and validity. Responsiveness and sensitivity to changes were also evaluated. In the development process of NPQ an initial list of 40 items was given to 60 patients with CRSwNP; the 27 most significant items were selected and converted into questions. The validation procedure involved 107 patients (mean age 52.9 ± 12.4). NPQ revealed a five-dimensional structure and high levels of internal consistency (Cronbach’s alpha 0.95). Convergent validity (Spearman’ coefficient r = 0.75; *p* < 0.01), discriminant validity (sensitivity to VAS score), and reliability in a sample of patients with a stable health status (Interclass Coefficient 0.882) were satisfactory. Responsiveness to clinical changes was accomplished. The minimal important difference was 7. NPQ is the first questionnaire for the assessment of HRQoL in CRSwNP. Our results demonstrate that the new tool is valid, reliable, and sensitive to individual changes.

## 1. Introduction

Chronic rhinosinusitis with nasal polyps (CRSwNP) is a chronic inflammatory disease of the paranasal sinuses affecting 2–4% of the general population [1]. It is the most severe subtype of CRS, characterized by symptoms often lasting for many years. Management of CRSwNP is difficult and recurrences are frequent, despite medical treatment and surgery. As a consequence, CRSwNP has a considerable impact on health-related quality of life (HRQoL). This expression refers to the impact of an illness and its therapy upon a patient, as perceived by the patient [2,3]. The burden of troublesome symptoms (nasal blockage, loss of smell, rhinorrhea, and sneezing), the presence of comorbid diseases (chronic rhinosinusitis, asthma, aspirin sensitivity), the necessity of long-term medical therapies, the need for surgical treatment, and the changes to habits and lifestyle all negatively impact physical, emotional and social aspects of daily life. Recently, novel biological treatments have been introduced in the management of more severe CRSwNP patients with promising results in terms of reduction of the global burden of CRSwNP [4].

Although the literature in this field is not rich, available data confirm the clinical findings [5]. Some studies have explored the subjective burden of CRSwNP by means of SF-36 [6], a generic measure that allows for the assessment of health status in patients and healthy subjects. Compared to the general population, patients with CRSwNP had worse scores in all SF-36 domains except for physical functioning [7]. The disease burden has also been detected by comparing CRSwNP with other chronic diseases, such as obstructive pulmonary diseases [8], asthma [9] and coronary artery disease [10]. No correlation was found between SF-36 scores and age, gender, nasal symptoms, CT scan, and polyp size [11]. The Sino-Nasal Outcome Test (SNOT-22) [12] a speciality-specific questionnaire that covers a broad range of rhinologic (12 questions) and general health issues (10 questions), represents the reference tool to assess symptoms and disease burden in patients with CRS. This well-validated tool was not specifically developed for CRSwNP, but its characteristics make it suitable for assessing the presence and the severity of sino-nasal disorders in clinical conditions: smell dysfunction [13], sino-nasal symptoms in cystic fibrosis [14], allergic rhinitis [15], sleep apnea [16], COPD [17], hereditary hemorrhagic telangiectasia [18], Wegener’s granulomatosis [19]. CRSwNP is the most bothersome phenotype of chronic rhinosinusitis. The clinical characteristics of the disease and the availability of new treatment options make it desirable to assess the impact of both CRSwNP and therapy by means of a specific HRQoL questionnaire.

The aim of the study was to develop and validate a specific questionnaire to assess HRQoL in patients affected by CRSwNP in order to complete the traditional assessment based on SNOT-22.

## 2. Materials and Methods

Consecutive patients who visited the Otorhinolaryngology and Personalized Medicine, Asthma and Allergy units at the Istituto Clinico Humanitas between September 2018 and May 2020 were invited to participate in the study.

The Ethics Committee of the Humanitas University (Milan) approved the study protocol (approval no. P.R. 1920). The protocol complies with the general principles of Good Clinical Practice and the Declaration of Helsinki as amended in Edinburgh in 2000. Participation was voluntary and anonymous, and informed consent was obtained from all patients before study entry.

The inclusion criteria were as follows: the presence of consistent symptoms and evidence of signs of CRSwNP at endoscopy; age ≥18 years; comprehension of spoken and written Italian language; availability and willingness to participate in the study.

Participants were excluded in cases of other ear–nose–throat disorders.

The development and validation of the new questionnaire occurred in two separate steps involving different groups of patients. The method used for the two phases is described in detail below.

### 2.1. Development Process

In order to make certain that the questionnaire included items appropriate and relevant for CRSwNP patients, items generation and selection was conducted on the basis of current guidelines [20,21,22]:

Item generation. The first step aimed to collect any potentially relevant and troublesome problems related to CRSwNP on the basis of the following sources: (i) literature review of the available HRQL questionnaires used with CRSwNP patients; (ii) a round-table with ENT specialists (2), pulmonologists (2) and a health psychologist; (iii) unstructured interviews with 10 adult outpatients with CRSwNP. This resultant list included practical, emotional, social and physical aspects of daily life that could be influenced by CRSwNP.

Item selection. The second step comprised an item importance ranking, in order to identify the most relevant problems related to CRSwNP. The questions found during the item generation procedure were randomly listed and individually administered to patients who were asked to indicate: (a) which of the items they experienced as consequence of CRSwNP; the response options were yes/no; (b) how relevant each of the identified items was, according to five-point response options indicating the degree of importance related to each item (1 = not important, 5 = very important)

In this phase, a sample of 60 consecutive outpatients with CRSwNP were accrued during a 2-month period. On the basis of collected data we calculated:the percentage of patients who identified each item as a consequence of CRSwNP (frequency range: 0–100);the mean importance attributed to each item (range: 0–4);the overall impact of each item, calculated as the product of the frequency and the mean importance divided by 100 (range: 0–4).

Selected items were converted to questions where patients had to indicate how much they had been troubled by each problem during the previous 2 weeks on a five-point Likert scale (1 = not at all, 5 = very much).

This format of the questionnaire was administered to a different group of patients for the validation process. Patients were selected using a convenience sampling method. The aim was to include almost 100 patients. We have called this the Nasal Polyposis Quality of Life (NPQ) questionnaire.

### 2.2. Validation Process

Patients were assessed twice with a 4-week interval between visits.

At both visits, a physician collected a complete and accurate medical history reporting the ongoing therapy and patients filled in the NPQ along with the following tools:-Visual analogue scale (VAS): patients were asked to indicate on a horizontal line measuring 10 cm the degree of CRSwNP severity, giving a score from 0 to 10. The score obtained can be divided into mild (VAS 0–3), moderate (VAS 3–7) and severe (VAS > 7) [1].-The SNOT-22 [12] encompasses 22 items, scored from 0 (meaning no problem reported) to 5 (as bad as it can be) scoring a maximum of 110 points. The higher the score the worse is the patient’s HRQoL. The SNOT-22 has been adapted and validated in several languages and it is now available also in Italian [23].

At Visit 2, patients completed the same questionnaires as at Visit 1 and a 7-point Global Rating Scale to assess any change in health status: 0 = “no change”, +1 (−1) = “small improvement (worsening)”, +2 (−2) = “moderate improvement (worsening), +3 (−3) = significant improvement (worsening)”.

The following psychometric properties of the NPQ were tested according to current guidelines [20,21,22]:-Construct validity was evaluated by means of factorial analysis; the principal component method with Varimax rotation was adopted.-Convergent validity was calculated by Spearman correlations to examine the relationships between the new questionnaire and an established measure (SNOT-22). Convergent validity is confirmed with correlations ranging from 0.4 to 0.8. Two instruments are considered too similar if the correlation is 0.8 or more (the tested instrument has no added value) [24].-Discriminant validity was evaluated comparing patients according to their VAS score by using ANOVA (Fischer’s test).-Internal consistency was estimated using Chronbach’s correlation coefficient on the extracted factors. Measures with reliability of 0.50–0.70 or greater were recommended for the purpose of comparing groups [25].-Reliability was evaluated by means of the Intraclass Correlation Coefficient (ICC) in the subsample of patients with a stable health status (GRS = 0). An ICC of >0.75 indicates excellent reproducibility while an ICC between 0.4 and 0.75 indicates a good reproducibility [25].-Responsiveness was assessed, analyzing the correlation between changes in the score of the new questionnaire and changes in GRS (GRS ≠ 0) and VAS by means of a non-parametric test (Spearman correlation coefficient).-Clinical significance was explored by assessing the minimal important difference (MID). The receiver operating characteristics (ROC) curve method was applied [26]. The entire cohort for one dichotomization point (i.e., ‘no change’ vs. ‘any improvement or deterioration’) was adopted.

The possible effect of age (Spearman’s correlation coefficient), gender, smoking habits and comorbid asthma (Fisher’s ANOVA) on patients’ answers was also tested. The frequency distribution of the answers was calculated to evaluate whether patients used the entire answer scale and whether all possible scores were obtained.

## 3. Results

### 3.1. Development Process

Item generation: the items generation phase resulted in a total of 54 items (Appendix A). The Authors performed a qualitative selection by eliminating 14 items that resulted as redundant (Table 1).

Item selection: sixty patients completed the development-phase questionnaire of 40 items. Most of these patients (63.3%) were female, and the mean (SD) age was 41.4 (8.3) years, ranging from 18 to 74 years.

On the basis of patients’ answers, items included in the questionnaire were those that scored highest in terms of impact. Where an arbitrary cut-off value of 1.5 was used for impact, 13 items were excluded. Table 1 summarizes the results of this first phase, indicating the 27 items selected according to the total importance.

### 3.2. Validation Process

We enrolled 107 subjects in the study. The mean age was 52.9 with a SD of 12.4; the majority were male (61.7%) and non-smokers (92.5%).

Comorbid asthma was found in 63 (58.9%) of patients.

Regarding atopy (as at least one allergen sensitization via skin prick test), 54 (50.5%) were found positive. Acetyl salicylic acid (ASA) intolerance, meaning patients reporting some kind of respiratory symptoms upon aspirin or any non-steroidal anti-inflammatory drugs (NSAIDS) intake, was found in 14 (13.1%) patients; 5 subjects out of 107 (4.7%) were affected by Samter’s triad.

-Construct validity: the factorial analysis with eigenvalue > 1 extracted five factors which explain up to 66.97% of the total variance. Items belonging to each factor are listed in Table 2.-Convergent validity: Spearman’s correlations between NPQ scores and SNOT-22 were significant (r = 0.75; *p* < 0.01).-Discriminant validity: the group of patients with VAS >7 had NPQ scores significantly higher than patients with VAS ≤ 7 (81.88 ± 21.02 vs. 61.4 ± 15.65, *p*-value < 0.001).-Internal consistency: Cronbach’s alpha coefficient value of 0.95 was obtained for the whole instrument, exceeding the minimum internal consistency standard of 0.70 recommended for group comparison.-Reliability: interclass coefficient (ICC) was 0.882, exceeding the cut-off of 0.75 indicating an excellent test reliability.-Responsiveness: the assessment of a subsample of 44 patients reporting an improvement or deterioration in health status (GRS 0) demonstrate that a significant Spearman correlation between the variation of NPQ total score between the two visits and the change in VAS score (0.628 *p* < 0.001) and in GRS (−0.528 *p* < 0.001) (Figure 1).-Clinical significance: results of the ROC analyses are presented in Table 3. A 7-point change in RAPP maximizes sensitivity, specificity, and the number of individuals correctly classified, identifying the MID.

With the use of a *t*-test, no significant difference was found in mean CRS-NP-QoL total score value comparing gender, comorbid asthma, atopy and ASA sensitivity. Smokers had a higher NPQ total score mean value compared to non-smokers (90.6 ± 20.1 vs. 74.3 ± 20.5, *p* = 0.03). No significant correlation was found between age and NPQ total score by the use of a linear regression analysis.

## 4. Discussion

HRQoL has become a crucial outcome in chronic conditions in order to capture the patient’s perspective about disease and treatment.

The availability of generic and specific questionnaires highlights that CRSwNP significantly affects patients’ HRQoL. SNOT-22, a widely used tool, also available in Italy, was not developed and validated to assess HRQoL impairment of patients suffering from CRSwNP. The DyNaChron is a self-reporting tool to assess nose and sinus functional symptoms and their consequences in patients with chronic nasal dysfunction [27]. This instrument was developed in French and then cross-culturally validated in Arabic-speaking Moroccan patients [28]. The questionnaire has excellent psychometric properties, but it consists of 78 items, which represents a disadvantage for use in our clinical practice. The Nasal Obstruction Symptom Evaluation (NOSE) [29] was developed to assess nasal obstruction, the most common rhinologic complaint in the ear, nose, and throat. It is well validated, brief, and easy to complete for the patient, but it evaluates only one symptom, which does not allow all aspects of HRQoL in CRSwNP to be collected. Recently, it has been shown that nasal polyposis might have a variable impact on HRQoL [30] and that patients with CRSwNP present a different HRQoL profile compared to those with CRSsNP [31].

To address this gap, we developed and validated a disease specific tool to detect HRQoL impairment in patients with CRSwNP, by following the established methodological guidelines and a recognized framework of questionnaire design [20,21,22]. The procedure we adopted provides evidence that the new instruments appropriately reflects the HRQoL of patients suffering from CRSwNP. In fact, the development process guarantees that the item selection has been determined by the patients on the basis of their experience.

The new questionnaire consists of 27 items that can provide a total score and/or five factorial scores. As expected, a moderate, significant correlation was obtained between NPQ and SNOT-22 (see in Appendix A).

Discriminant validity was demonstrated through the tool’s ability to discriminate between groups defined according to the VAS.

NPQ was shown to be an internally reliable tool as indicated by very high Cronbach α coefficients (0.95). It was also a reliable questionnaire as supported by satisfactory ICC in stable patients (0.882). High responsiveness to changes were confirmed by a significant correlation between the change of NPQ. Total score between the two visits and the change in VAS score and in GRS. The ROC analysis indicates that a seven-point score is the smallest change that patients perceive as an improvement or deterioration. The use of the entire cohort in ROC analysis, rather than just the two groups adjacent to the dichotomization point, was found to maximize sensitivity and specificity, increasing confidence in the point estimates [26].

The new questionnaire has several advantages when compared to other validated tools: it was specifically built to detect specific problems and dimensions that cannot be captured by the other available tools (i.e., worrying about long-term drug efficacy and possibility of surgery, embarrassment); and it is simple to complete and to score. Moreover, the NPQ possesses the necessary psychometric properties that make it a valid instrument; the cutoff MID makes it easy to determine the clinical significance of the results and changes over time, hence, making it suitable for evaluating the effects of treatment. Moreover, answers were not influenced by socio-demographic characteristics, thus enabling the NPQ to be used regardless of the patient’s sex, age and education.

Because of these characteristics, NPQ is appealing as an instrument to assess the patient experience of CRSwNP. It is also potentially useful to monitor the impact of both disease and treatment from the patient’s perspective owing to its satisfactory responsiveness to changes.

Although we reached the primary aim of our study by providing evidence to support the validity, reliability, and responsiveness of NPQ, our findings should be considered in the light of the following potential limitations.

First, the generalizability of the results should be limited because the sample was non-randomized and the patients were enrolled in one specialist center. Second, no objective measures of disease control, severity and extent, besides the patient’s reported outcomes, were adopted to determine the reliability and sensitivity to change. Third, the acceptability of the new tool for both patients and physicians has not been evaluated. However, these limitations may be addressed through further studies.

## 5. Conclusions

In conclusion, NPQ is the first questionnaire for the assessment of HRQoL in CRSwNP. It is valid, reliable, and sensitive to individual changes. It is able to detect the specific burden of CRSwNP on HRQoL. This tool should yield data to improve our ability to effectively monitor the burden of disease and treatment on patients with CRSwNP.

## Figures and Tables

**Figure 1 healthcare-10-00253-f001:**
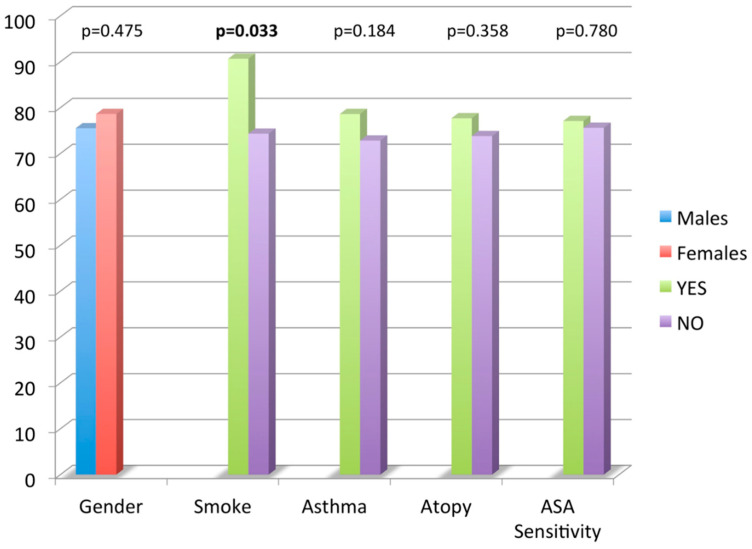
Nasal Polyposis Quality of Life (NPQ) questionnaire total score mean values according to age, smoking habits, asthma, atopy and acetyl salicylic acid (ASA) sensitivity.

**Table 1 healthcare-10-00253-t001:** Development process: results of item reduction.

N Item	Item	Frequency (0–100)	Mean Importance (0–4)	Overall Impact (0–4)
**1**	**Difficulties sleeping**	**73.33**	**2.81**	**2.06**
2	Having to spend money/treatment costs	65	2.26	1.47
**3**	**A dry throat/mouth**	**76.67**	**2.67**	**2.05**
**4**	**Inability to fully carry out sporting activities**	**63.33**	**2.66**	**1.69**
**5**	**Bad breath**	**65**	**2.72**	**1.76**
6	Inability to fully carry out routine daily activities	55	2.67	1.46
7	Waking up during the night to drink	60	2.39	1.43
**8**	**A bad taste in your mouth**	**60**	**2.67**	**1.60**
**9**	**Difficulty tasting food and wine**	**81.67**	**3.41**	**2.78**
**10**	**Feeling irritable**	**70**	**2.95**	**2.06**
**11**	**Difficulty concentrating**	**58.33**	**2.94**	**1.71**
**12**	**Feeling tired**	**66.67**	**3.1**	**2.07**
**13**	**A poor sense of smell**	**86.67**	**3.81**	**3.38**
**14**	**Feeling uncomfortable around other people**	**60**	**2.89**	**1.73**
**15**	**Feeling embarassed due to symptoms**	**63.33**	**2.61**	**1.65**
**16**	**A kneaded mouth** **Furry tongue?**	**63.33**	**2.71**	**1.71**
**17**	**Being worried**	**73.33**	**2.79**	**2.04**
18	Feeling anxious	50	2.26	1.13
**19**	**Feeling embarassed/socially in your social life**	**63.33**	**2.42**	**1.53**
20	Dark circles under your eyes	53.33	2.34	1.24
21	A Swollen face	55	1.01	0.56
22	The need for computed tomography (CT) scans	53.33	1.91	1.02
23	Hearing problems	50	2.5	1.25
**24**	**Being worried about the side effects of medication**	**70**	**2.8**	**1.96**
**25**	**The need for possibile surgery**	**81.67**	**2.89**	**2.36**
26	The need for frequent medical check-ups	50	2.2	1.10
**27**	**Feeling stressed**	**65**	**2.49**	**1.62**
**28**	**Not feeling in control of the disease**	**71.67**	**3.37**	**2.41**
**29**	**A nasal voice**	**78.33**	**2.7**	**2.11**
**30**	**Snoring at night**	**76.67**	**2.67**	**1.71**
31	The need to undergo invasive clinical examinations	58.33	2.23	1.30
32	Sexual activity is compromised	48.33	2.13	1.03
**33**	**Feeling worried about long term drug efficacy**	**69.74**	**3.01**	**2.10**
34	Difficulty kissing	50	2.23	1.11
**35**	**Difficulties managing symptoms**	**85**	**2.45**	**2.08**
**36**	**Feeling worried that the problem will recur**	**85**	**3.21**	**2.73**
**37**	**Feeling worried that you may not notice personal unpleasant body odours (e.g., when you sweat)**	**75**	**3.22**	**2.41**
38	Facial pain	45	2.33	1.05
**39**	**Headaches**	**71.67**	**2.67**	**1.91**
**40**	**Inability to do as much as you would like to do**	**57.89**	**3.06**	**1.77**

Bold faces indicate high importance items (overall impact ≥ 1.5).

**Table 2 healthcare-10-00253-t002:** Factors identified using principal components analysis on full data set: 1—Daily life impact; 2—Mouth problems; 3—Embarrassment; 4—Treatment impact; 5—Loss of smell.

Item	Factors				
	1	2	3	4	5
**Sleeping disorders**	**0.520**	0.341	0.240	0.311	0.270
**A dry throat/mouth**	**0.570**	0.298	0.311	0.090	0.274
**Inability to fully carry out sporting activities**	**0.602**	0.378	−0.111	0.159	0.275
**Bad breath**	0.077	**0.731**	0.099	0.028	0.067
**Difficulty tasting food and wine**	0.165	0.124	0.129	−0.008	**0.801**
**Being irritable**	**0.583**	0.511	0.227	0.266	0.061
**Being worried about the side effects of medication**	0.341	−0.056	0.394	**0.656**	−0.051
**Feeling embarassed socially/in your social life**	0.492	0.069	**0.656**	0.172	0.145
**A nasal voice**	0.138	0.198	**0.720**	0.021	0.136
**Being worried about the disease**	**0.558**	0.111	0.358	0.446	0.046
**Not feeling in control of the disease**	**0.721**	−0.025	0.234	0.257	0.278
**Feeling worried that you may not notice personal unpleasant body odours (e.g., when you sweat)**	0.325	0.218	0.217	0.111	**0.593**
**Headaches**	0.046	0.443	**0.497**	0.030	0.199
**Feeling worried that the problem will recur**	**0.658**	−0.080	0.167	0.356	0.252
**Being worried about possible surgery**	0.092	0.129	−0.053	**0.792**	0.022
**Being stressed**	0.077	**0.562**	−0.126	0.304	0.204
**Snoring**	**0.691**	0.372	0.170	0.407	0.027
**Difficulty concentrating**	**0.693**	0.315	0.263	−0.002	0.153
**A poor sense of smell**	0.115	0.092	0.105	0.082	**0.836**
**Feeling embarassed due to the symptoms**	0.490	0.078	**0.493**	0.380	0.261
**Having a bad taste in your mouth**	0.251	**0.723**	0.433	−0.069	0.036
**Kneaded mouth** **Furry tongue**	0.339	**0.644**	0.404	0.091	0.215
**Feeling tired**	**0.691**	0.506	0.012	0.227	−0.002
**Being worried by long term drug efficacy**	0.326	0.172	0.098	**0.662**	0.171
**Feeling uncomfortable with other people**	**0.554**	0.059	0.529	0.281	0.181
**Having difficulty in controlling symptoms**	**0.761**	0.004	0.181	0.212	0.201
**Not performing well**	**0.847**	0.218	0.141	0.005	0.031

Bold typeface shows the component upon which each item loaded most highly.

**Table 3 healthcare-10-00253-t003:** The minimal important difference (MID) of chronic rhinosinusitis with nasal polyps impact on quality of life (CRSwNP-QoL) obtained with the receiver operating characteristics (ROC) analysis with different cutoff.

Cutoff ≥	Sensitivity (%)	1-Specificity (%)
11	0.77	0.69
9	0.80	0.69
7 *	0.83	0.63
5	0.83	0.44
3	0.87	0.06

* cutoff point chosen.

## Data Availability

The data presented in this study are available on request from the corresponding author. The data are not publicly available due to privacy of the patients.

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
