# Peer review of "Nasal Polyposis Quality of Life (NPQ): Development and Validation of the First Specific Quality of Life Questionnaire for Chronic Rhinosinusitis with Nasal Polyps"

_healthcare, 2022, doi:10.3390/healthcare10020253_

Round 1
Reviewer 1 Report
This is a well thought out study meant to add a significant component of information to the literature that would be very useful to the rhinologist. A few thoughts and comments;
1) It appears that the study was conducted in Italian - will an English, and other language version be available, especially since it is being published in an English journal? Will the NPQ questions be published in English in the manuscript?
2) The english language structure needs to be worked on quite a bit throughout the manuscript.
3) Since all these patients were CRSwNP patients, it would be good to add a correlation between your new NPQ and other objective measures ie CT scan findings and endoscopic findings.
Author Response
- It appears that the study was conducted in Italian - will an English, and other language version be available, especially since it is being published in an English journal? Will the NPQ questions be published in English in the manuscript?
Authors’ response: Thank you for this comment. The questionnaire has beeen developed and validated in Italian. The questions have been translated into English in the manuscript. The authors will provide the Italian version to those who request it.Transcultural adaptation and validation of the NPQ in other languages is is one of our next research goals.
- The english language structure needs to be worked on quite a bit throughout the manuscript.
Authors’ response: As Reviewer recommends, the manuscript has been revised by an editing service. The changes constitute grammatical improvements that do not fundamentally change the meaning or content.
- Since all these patients were CRSwNP patients, it would be good to add a correlation between your new NPQ and other objective measures ie CT scan findings and endoscopic findings.
The study protocol has been designed to assess, as clinical objective measure, the Visual analogue scale (VAS). This method has been recommended by the EPOS2020: European Position Paper on Rhinosinusitis and Nasal Polyps 2020. Validation is an iterative process, and this study constitutes the first phase in that process. Further work is planned to examine the correlation between your new NPQ and other objective measures ie CT scan findings and endoscopic findings
Reviewer 2 Report
This is a novel study for developing a new questionnaire for QOL in CRSwNP patients. However, much revision is needed for publication.
Abstract needs revision. Is the abstract written according to the journal’s guideline ? The description in the Materials and Methods is too vague. English proofreading is also required. The full manuscript also needs to be revised for English. Since the manuscript is a little bit distracting, please re-organize the paragraphs throughout the manuscript.
In line 85, how was CRSwNP diagnosed ? By endoscopy or CT ?
Are the patients in the development process and validation procedure different ? Please mention this in the manuscript.
Supplementary Table should be mentioned in the manuscript.
In line 208, Figure legend should be revised.
So, where is the final 27 questionnaire you developed ? Please provide the final NPQ questionnaire for the readers to use in their practice. Is there any cut-off value ?
Why are the items in Table 1 and Table 2 different ?
In Discussion, how is your questionnaire different from other reported questionnaires such as NOSE scale ? Limitation of the study should include the extent of the CRSwNP is diverse among the patients included.
Author Response
Abstract needs revision. Is the abstract written according to the journal’s guideline ? The description in the Materials and Methods is too vague. English proofreading is also required. The full manuscript also needs to be revised for English. Since the manuscript is a little bit distracting, please re-organize the paragraphs throughout the manuscript.
Authors’ response: Thank you for this comment, according to which we have reviewed the abstract. The manuscript has been revised by an editing service.
In line 85, how was CRSwNP diagnosed? By endoscopy or CT ?
Authors’ response: all the patients were diagnosed according to the presence of consistent sympotms and evidence of signs of CRSwNP at endoscopy. We added this information in the revised manuscript (line 155).
Are the patients in the development process and validation procedure different ? Please mention this in the manuscript.
Authors’ response: The development and validation of the new questionnaire occurred in two separate steps involving different groups of patients. It has been better specified in the method and in the results sections.
Supplementary Table should be mentioned in the manuscript.
Authors’ response: Thank you for the comment. Supplementary table has been mentioned in the Results section
In line 208, Figure legend should be revised.
Authors’ response: Figure legend has been revised
So, where is the final 27 questionnaire you developed ? Please provide the final NPQ questionnaire for the readers to use in their practice. Is there any cut-off value ?
Authors’ response: Table 2 includes all the 27 items. The questionnaire is now validated only in Italian. The list of items reported in Table 2 is a simple English translation of the Italian questions. The authors will provide the Italian version to those who request it.
Why are the items in Table 1 and Table 2 different ?
Authors’ response: Table 1 reports the result of the development process (27 items selected and 13 items deleted). Table 2 reports items belonging to each factor.
In Discussion, how is your questionnaire different from other reported questionnaires such as NOSE scale ? Limitation of the study should include the extent of the CRSwNP is diverse among the patients included.
Authors’ response: The specificity of NPQ has been better explained in the discussion. We did not mention the NOSE in our manuscript because, althogh its a well validated tool, in does not provide an assessment on HRQoL of CRSwNP patients, but it focuses on a symptom, nasal obstruction.We added the suggested limitation into the Discussion.
Reviewer 3 Report
The authors present in this article a new quality of life questionnaire specific to CRSwNP. Some parts are generally well written, however there are many points deserve to be noted.
The introduction is generally well written, with up-to-date references. We could regret the fact of not talking about biotherapies for which an assessment of the quality of life is essential for follow-up. The Dynachron questionnaire, developed by Jankowski, should be cited because it takes into account many points of the questionnaire used by the authors.
The main negative point of this study is that the questionnaire is used in Italian by the authors and presented in the article in English. The validation of the questionnaire should first be validated in Italian, then subsequently translated and validated in English. This point is the main flaw in this study and therefore makes publication in this state impossible.
The items presented in the questionnaire are poorly developed and deserve to be further developed.
In table 2, the explanation of factor 5 is missing/
The methodology is well explained but very heavy. Simplification would be desirable.
The value of this questionnaire compared to SNOT-22, a benchmark score in the CRSwNP, is not obvious to me.
Author Response
The introduction is generally well written, with up-to-date references. We could regret the fact of not talking about biotherapies for which an assessment of the quality of life is essential for follow-up. The Dynachron questionnaire, developed by Jankowski, should be cited because it takes into account many points of the questionnaire used by the authors.
Authors’ response: Thank you very much. We have revised the introduction and the discussion according to your suggestions.
The main negative point of this study is that the questionnaire is used in Italian by the authors and presented in the article in English. The validation of the questionnaire should first be validated in Italian, then subsequently translated and validated in English. This point is the main flaw in this study and therefore makes publication in this state impossible.
Authors’ response: Thank you for this comment. The questionnaire has beeen developed and validated in Italian. The questions have been simply translated into English in the manuscript. The authors will provide the Italian validated version to those who request it. Transcultural adaptation and validation of the NPQ in other languages is is one of our next research goals.
The items presented in the questionnaire are poorly developed and deserve to be further developed.
Authors’ response: The items are intentionally simple, to increase their comprehensibility. The list of items is introduced by this sentence “The following questionnaire will evaluate the impact Chronic Rhinosinusitis with Nasal Polyps (CRSwNP) in daily life. Please indicate how much you have been bothered during the last 2 weeks by…”
In table 2, the explanation of factor 5 is missing
Authors’ response: The explanation has been corrected
The methodology is well explained but very heavy. Simplification would be desirable.
Authors’ response: The validation is a complex procedure. We have tried to better explane to make the process more readible
The value of this questionnaire compared to SNOT-22, a benchmark score in the CRSwNP, is not obvious to me.
Authors’ response: The specific caracheristics of NPQ and its differences from other well validate questionnares have been better explained in the discussion.
Round 2
Reviewer 2 Report
All the worries and suggestions made at the first review are now solved and well documented. Thank you for your hard work.
Reviewer 3 Report
The authors responded to all comments and made the requested changes